# *PEA15* loss of function and defective cerebral development in the domestic cat

Emily C. Graff [1,2,3☯]*, J. Nicholas Cochran [4☯], Christopher B. Kaelin [4], Kenneth Day[4], Heather L. Gray-Edwards [2,5], Rie Watanabe [1], Jey W. Koehler[1,3], Rebecca A. Falgoust [2], Jeremy W. Prokop[4], Richard M. Myers [4], Nancy R. Cox[1,2†], Gregory S. Barsh [4]*, Douglas R. Martin[2,3,5]*, 99 Lives Consortium[¶]

**1** Department of Pathobiology, College of Veterinary Medicine, Auburn University, Auburn, United States of America, **2** Scott-Ritchey Research Center, College of Veterinary Medicine, Auburn University, Auburn, Alabama, United States of America, **3** Center for Neuroscience Initiative, Auburn University, Auburn, AL, United States of America Alabama, United States of America, **4** HudsonAlpha Institute for Biotechnology, Huntsville, AL, United States of America, **5** Department of Anatomy Physiology and Pharmacology, College of Veterinary Medicine, Auburn University, Auburn, Alabama, United States of America

☯ These authors contributed equally to this work.
† Deceased.
¶ Membership of the 99 Lives Consortium is listed in the Acknowledgments.
* ecg0001@auburn.edu (ECG); gbarsh@hudsonalpha.org (GSB); martidr@auburn.edu (DRM)

**Data Availability Statement:** All sequencing data are publicly available at SRA Project PRJNA495843. Descriptors for each sample are

## Abstract

Cerebral cortical size and organization are critical features of neurodevelopment and human evolution, for which genetic investigation in model organisms can provide insight into developmental mechanisms and the causes of cerebral malformations. However, some abnormalities in cerebral cortical proliferation and folding are challenging to study in laboratory mice due to the absence of gyri and sulci in rodents. We report an autosomal recessive allele in domestic cats associated with impaired cerebral cortical expansion and folding, giving rise to a smooth, lissencephalic brain, and that appears to be caused by homozygosity for a frameshift in *PEA15* (phosphoprotein expressed in astrocytes-15). Notably, previous studies of a *Pea15* targeted mutation in mice did not reveal structural brain abnormalities. Affected cats, however, present with a non-progressive hypermetric gait and tremors, develop dissociative behavioral defects and aggression with age, and exhibit profound malformation of the cerebrum, with a 45% average decrease in overall brain weight, and reduction or absence of the ectosylvian, sylvian and anterior cingulate gyrus. Histologically, the cerebral cortical layers are disorganized, there is substantial loss of white matter in tracts such as the corona radiata and internal capsule, but the cerebellum is relatively spared. RNA-seq and immunohistochemical analysis reveal astrocytosis. Fibroblasts cultured from affected cats exhibit increased TNFα-mediated apoptosis, and increased FGFb-induced proliferation, consistent with previous studies implicating PEA15 as an intracellular adapter protein, and suggesting an underlying pathophysiology in which increased death of neurons accompanied by increased proliferation of astrocytes gives rise to abnormal organization of neuronal layers and loss of white matter. Taken together, our work points to a new role for *PEA15* in development of a complex cerebral cortex that is only apparent in gyrencephalic species.

included at SRA, and also in the supplemental table (S1 Table).

**Funding:** The work was partially supported by a grant from the National Institutes of Health to GSB (AR067925), by the HudsonAlpha Institute of Biotechnology, and by Auburn University. The funders had no role in study design, data collection and analysis, decision to publish, or preparation of the manuscript.

**Competing interests:** The authors have declared that no competing interests exist.

## Summary

Gyrification is the neurodevelopmental process in certain mammalian species during which the cerebral cortex expands and folds resulting in the classic wrinkled appearance of the brain. Abnormalities in this process underlie many congenital malformations of the brain. However, unlike many other human malformations, genetic insight into gyrification is not possible in laboratory mice because rodents have a lissencephalic or smooth cerebral cortex. We identified a pathogenic variant in domestic cats that likely causes failure of the cerebral cortex to expand and fold properly, and discovered that the pathogenic variant impairs production of a protein, PEA15 (phosphoprotein expressed in astrocytes-15), involved in intracellular signaling. Affected cats have profound abnormalities in brain development, with minimal changes in their superficial behavior and neurologic function. Additional studies of tissue and cultured cells from affected animals suggest a pathophysiologic mechanism in which increased death of neurons accompanied by increased cell division of astrocytes gives rise to abnormal organization of neuronal layers and loss of white matter. These results provide new insight into a developmental process that is unique to animals with gyrencephalic brains.

## Introduction

Cerebral dysgenesis, or abnormal development of the telencephalon, encompasses a large number of malformations of cortical development including cortical dysplasia, microcephaly, heterotopia, schizencephaly, and polymicrogyria [1]. Clinical presentation of patients with cerebral dysgenesis can range from intellectual disability to severe epilepsy and neural tube defects [2]. Mendelian causes of cerebral dysgenesis in humans includes loss of function variants in *WDR62*, *NDE1*, *DYNC1H1*, *KIF5C*, *KIF2A*, and *TUBG1* and related genes [3]. The vast majority of genes that regulate advanced cerebral cortical development have been discovered via forward genetic approaches in humans.

Gyrification refers to the process by which the cerebral cortex expands and folds. Overall size and gyrification of the cerebral cortex varies significantly between species [4], and increased cortical mass, cortical gyrification, and complex lamination within the cerebral cortex are traits that are associated with cognitive ability [5, 6]. Appropriate *in vivo* models of gyrification are limited as most laboratory animals exhibit minimal gyrification, and rodents (the most commonly used laboratory models) are lissencephalic [4]. However, the few studies that do exist in gyrencephalic models provide important insights into mechanisms of gyrification [7, 8], and gyrification studies are being conducted in relevant gyrencephalic species such as cats, sheep, and dogs [4]. Cats have prominent gyrification and are commonly used as a model for numerous neurologic diseases [9, 10].

Here we report that a loss of function variant in *PEA15* (XM_023247767.1:c.176delA, XP_023103535.1:p.(Asn59fs), felCat9 chrF1:66768323 GT -> G) is likely responsible for a form of cerebral dysgenesis in the domestic cat, characterized by microcephaly and polymicrogyria. Characterization of the pathophysiological and neurodevelopmental consequences of *PEA15* deficiency offers insight into its essential role in gyrification and cortical development.

## Results

### Cerebral dysgenesis underlies an inherited neurodevelopmental abnormality in cats

An autosomal recessive, neurodevelopmental abnormality spontaneously arose in a domestic cat closed breeding colony at Auburn University that was established to study lysosomal

storage diseases, and in which pathogenic variants are segregating for GM2 gangliosidosis variant AB (*GM2A*) [11], and mucopolysaccharidosis VI (*MPSVI*) [12]. The cerebral dysgenesis phenotype segregated independently from both of the known pathogenic variants in the colony (S1 Table) and is clinically distinct from the lysosomal storage disease phenotypes, which lead to a progressive neurologic degeneration that is typically fatal prior to one year of age. Evaluation of 123 cats from the two breeding colonies identified 25 cats that were phenotypically affected with cerebral dysgenesis (S1 Fig). Of the cats affected with cerebral dysgenesis, 6 were heterozygous and 3 were homozygous for the *MPSVI* mutant allele (16 were homozygous reference). while 1 was heterozygous and 3 were homozygous for the *GM2A* mutant allele (21 were homozygous reference). None of the cats with cerebral dysgenesis were doubly homozygous for *MPSVI* and *GM2A* mutant alleles, and expressivity of the cerebral dysgenesis phenotype was not affected by carrier status of either *MPSVI* or *GM2A*. Twelve of the 25 cats with cerebral dysgenesis were homozygous reference for both *MPSVI* and *GM2A* mutant alleles.

Animals with cerebral dysgenesis exhibit spastic tetraparesis and ataxia first apparent around 3–4 weeks of age as they begin to walk. As the animals grow, spasticity and ataxia partially resolve, stabilizing by 6–9 months of age. At approximately 1.5 years, affected cats develop sensory abnormalities and often become aggressive. Sensory abnormalities manifest as stargazing and fly-biting: staring into empty space, and attacking or biting with no stimulus present, respectively. Aggressive behaviors were often erratic, unpredictable and included unprovoked attacks on long-term cage mates and caretakers. Some affected animals had seizures, although abnormal baseline EEG tracings were not observed. There were no deviations from reference intervals in complete blood count, serum biochemistry or urinalysis of affected cats, indicating that apart from the severe neurologic changes there was no additional systemic disease. Cerebrospinal fluid in carriers and affected cats showed no abnormalities or evidence of central nervous system inflammation (S2 Table). In addition, cerebrospinal fluid enzyme activity for markers of inflammation and neuronal cell damage were not significantly different from unaffected age-matched cats (S2 Fig). Taken together, these findings suggest that affected adult cats had a stable, non-progressive neurologic disease.

At necropsy, affected juvenile and adult cats exhibited generalized microcephaly and polymicrogyria with focal lissencephaly and regional gyral variability (Fig 1A–1C). The most severely affected areas (frontoparietal) often had a cobblestone appearance. The average brain weight of affected cats was 4.3 grams per kilogram of body weight, while for unaffected and carrier cats average brain weight was 7.8 grams per kilogram, indicating a 45% decrease in brain mass (Fig 1D). In contrast to the significant abnormalities in the cerebral cortex, the size and structure of the cerebellum was normal with no vermal or hemispheric hypoplasia, dysplasia or agenesis.

Brains from affected cats aged 1–8 months were evaluated histologically in comparison to aged-matched controls (Fig 1E). Overall, affected cats exhibited variable thinning of the cerebral cortex, especially in dorsal and lateral regions, and disorganization of cortical layers. In severely affected areas, cortical neurons were present in an undulating laminar band reminiscent of gyri, but sulci were largely absent and gyral folds were irregular in size, location, distribution, and orientation. White matter of the corona radiata and internal capsule was markedly decreased in volume, but the corpus callosum was generally spared. Basal nuclei appeared normally organized, as did olfactory tubercles, olfaction tracts and thalamus, although all areas were smaller than age-matched controls. Recapitulating the impressions garnered at necropsy, the cerebellum was normally organized with all layers including an age-appropriate external granule layer in cats younger than 3–4 months.

Magnetic resonance imaging (MRI) showed similar gyrification abnormalities as observed grossly in necropsy samples with flattening of the parietal and temporal lobes (S 2).

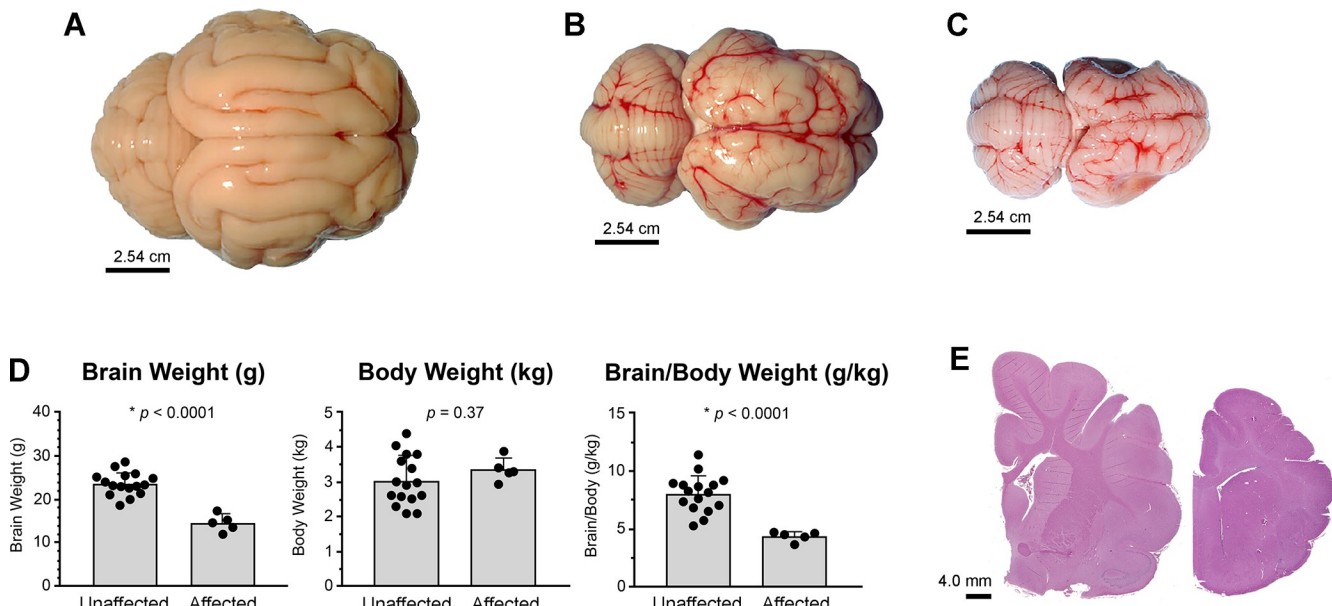

**Fig 1. Affected cats have marked microcephaly with polymicrogyria.** Images of whole brain from (**A**) adult unaffected (+/+), and (**B**) adult and (**C**) juvenile affected (-/-) cats. Affected cats have dramatically decreased cerebral cortex size with normal formation of the cerebellum. (**D**) Brain weights of affected cats are significantly decreased with or without normalization to body weight, which is similar to unaffected cats. (**E**) Representative sections (left image, normal; right image, affected) from the region of the parietal cortex have gyrification defects characterized by shallow sulci and fusion of small gyri consistent with polymicrogyria, as well as abnormal white matter of the corona radiata and internal capsule.

Additionally, MRI suggested that decreased brain volume was due primarily to decreased white matter. The most dramatic changes noted on MRI were located at the perisylvian, ecto-sylvian and cingulate gyri (Fig 2A–2C), where there was serve attenuation of gyral formation and decreased white matter, particularly in the anterior region (Fig 2C).

## PEA15 loss of function as the most likely Mendelian cause of cerebral dysgenesis

To identify the underlying genetic cause of cerebral dysgenesis, we initially carried out whole genome sequencing (WGS) and RNA-seq-based genotyping of eight affected animals and six obligate carriers. Because the Auburn research colony has been closed for nearly 50 years, and because the cerebral dysgenesis phenotype has not been described previously, we hypothesized that it was caused by homozygosity for a private variant. Applying zygosity-based filtering criteria under the model of complete penetrance of an autosomal recessive trait, we identified all variants that were homozygous in eight affected animals and heterozygous in the six obligate carriers, and observed a cluster of variants that was significantly enriched ($p = 0.006$, chi-square vs. random genome-wide distribution) in a 5 Mb region towards the end of chromosome F1 (Fig 3A).

To confirm and further fine-map the region, we carried out amplicon-based genotype-by-sequencing on an additional 91 cats (Fig 3, S1 Table), using 26 variants across an ~70 Mb interval that contains the candidate region on chromosome F1, 4 additional variants that span the remainder of chromosome F1, and 20 variants from other chromosomes. All affected cats, but no unaffected cats, were homozygous for a 1.3 Mb haplotype on chromosome F1 (Fig 3B) in which the peak LOD score was 10.1 (Fig 3C, S3 Table); all variants on chromosomes other than F1 exhibited LOD scores < 1.8 (S3 Table), and the variants used for initial zygosity filtering criteria are presented in S4 Table.

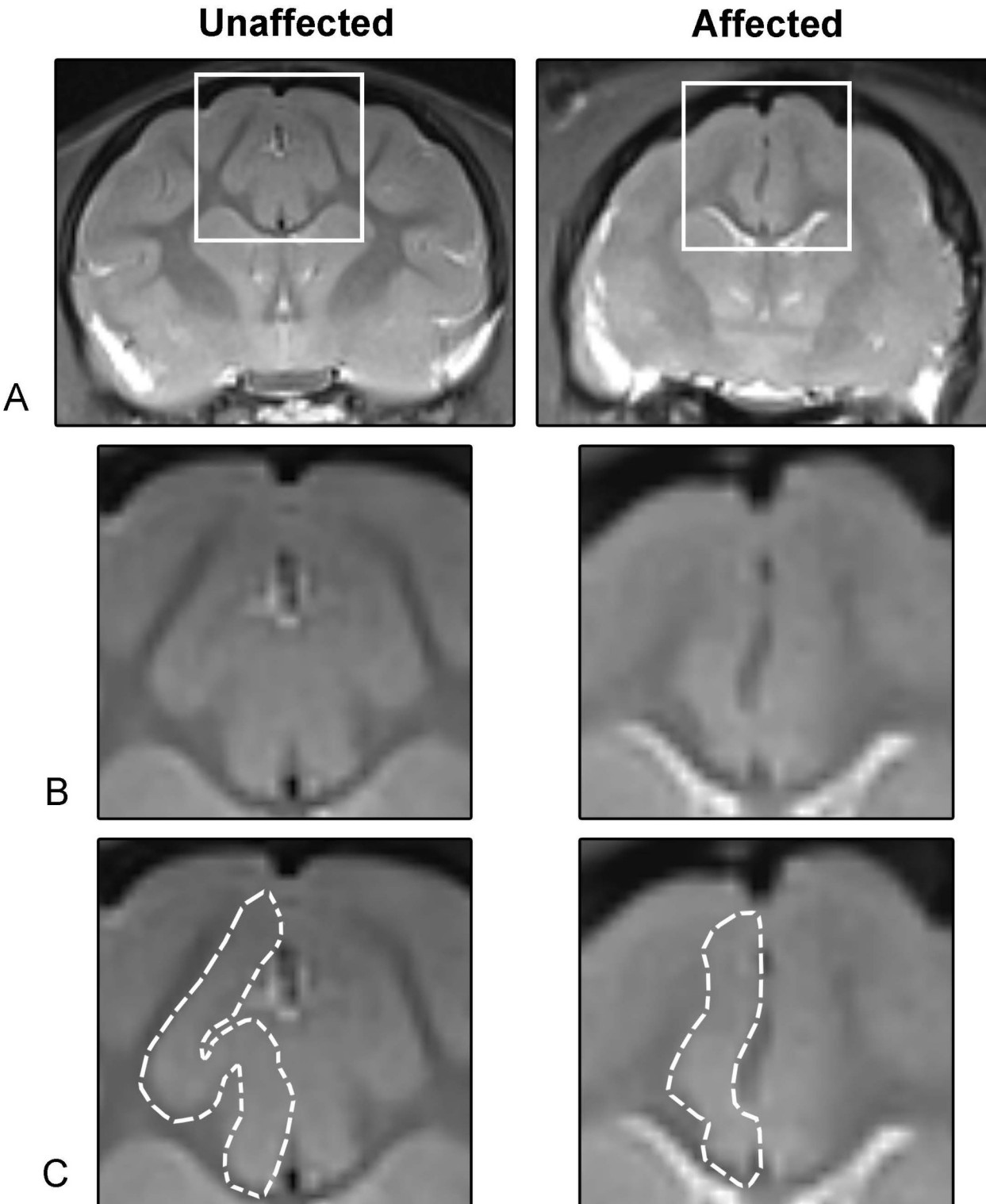

**Fig 2. Changes in MRI are consistent with microcephaly and attenuation of gyral formation.** (**A**) Selected images from the frontoparietal region demonstrate marked attenuation and loss of gyral formation and white matter. Note the blurring of gray and white matter boundaries, especially apparent in the corona radiata. (**B**) Magnified region within the white box highlights the severe attenuation of the (**C**) anterior cingulate gyrus, outlined in white.

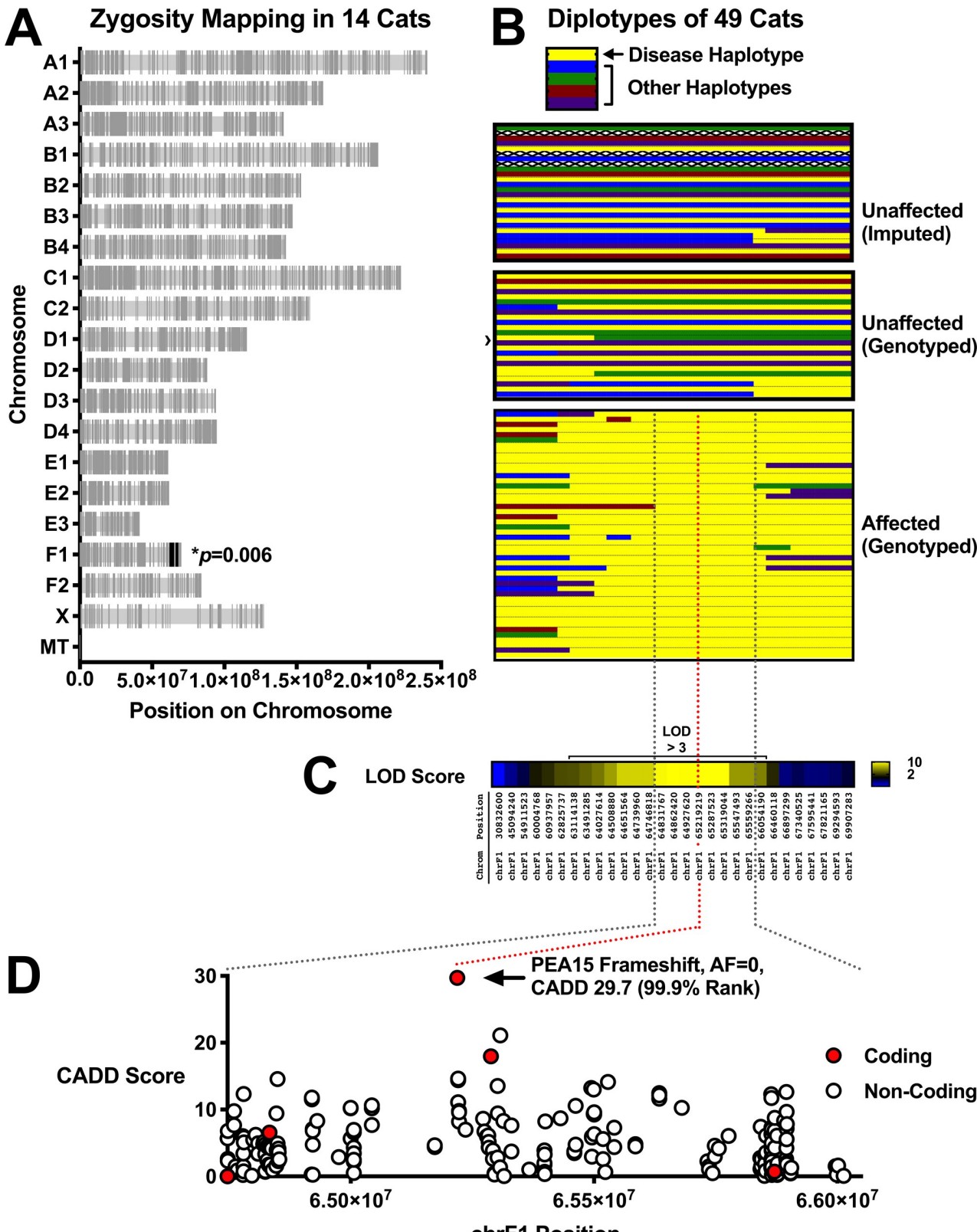

**Fig 3. Zygosity mapping, linkage, and haplotype analysis identifies a frameshift in *PEA15* as the cause of cerebral dysgenesis.** (A) Zygosity mapping, identifying all variants that are homozygous in 8 affected animals and heterozygous in 6 obligate carriers. Variants cluster in a region on the

distal end of chromosome F1. (**B**) Diplotypes of 49 cats according to disease status as indicated. 13 diplotypes were imputed from progeny: the top 4 diplotypes are founders (note uncertain haplotype, denoted by X's), the next 5 diplotypes are for the next generation after the founders, and 4 other cats throughout the pedigree were imputed because a sample was not available. In the unaffected genotyped block, the cat indicated with a › is a cat that has 2 normal diplotypes but is present in the analysis because it was bred with a cat homozygous for the disease diplotype. All affected animals are homozygous for a 1.3 Mb region (dashed black lines) (**C**) Linkage analysis confirms that the 1.3 Mb region on chromosome F1 identified by zygosity and haplotype analysis cosegregates with cerebral dysgenesis (coordinates given according to Felcat8). (**D**) CADD scores for all variants in the 1.3 Mb critical region that are homozygous in affected cats and at less than 5% allele frequency in the 99 Lives dataset. Only nine coding variants are present (see detail in **Table 1**). Six of the coding variants are synonymous. Two variants are missense in *LY9* and *CD48* (neither of which is expressed in brain). The synonymous variant in *SLAMF1* is listed in Table 1 but not plotted here because its CADD score is inflated as it is missense in human, but synonymous in cat. Two other coding variants are also not plotted because they did not lift over to human, but are presented in Table 1. Finally, the two coding variants in the lower left corner are nearby variants in *LY9* that cannot be distinguished on this plot because they both have a CADD score near 0. The final coding variant in the region is a frameshift in *PEA15*, which is highly expressed in brain. The variant is predicted to be highly damaging by CADD.

Within the 1.3 Mb critical region, there are 2289 variants, of which 337 are private (not present in the 99 Lives dataset (an allele frequency database for cats described elsewhere [13]); four in protein-coding sequence and 333 that are non-coding. An additional 150 variants (five coding, 145 non-coding) are present in 99 Lives at an allele frequency < 5%. We evaluated potential deleteriousness of both sets of variants (487) with combined annotation dependent depletion (CADD) scores (Fig 3D). One variant, a single nucleotide deletion in *PEA15* (XM_023247767.1:c.176delA, XP_023103535.1:p.(Asn59fs)), stood out as the strongest candidate. *PEA15* is highly expressed in the brain, the candidate variant in affected animals is not found in the 99Lives dataset, has the highest CADD score in the critical interval (29.7), predicts a frameshift and early truncation, and likely leads to nonsense-mediated decay as described below. Table 1 depicts all nine coding variants in the critical interval together with a non-coding variant that has the next highest CADD score, and illustrates that none lie in genes that are expressed in the brain with one exception: a synonymous variant in ATP1A2 that has an allele frequency of 4.4% in the 99Lives dataset.

In addition to considering single nucleotide and small indel variants, we surveyed the critical region for copy number and structural variants (Methods), and did not detect any changes consistent with the previously observed zygosity pattern. Furthermore, no assembly or coverage gaps were present in the 1.3 Mb candidate interval, and we found no evidence for structural variants based on discordant or clipped reads (S3 Fig).

**Table 1. Key sequence variants (coding or CADD>15) in the 1.3 Mb critical interval.** The table contains all coding variants in the interval with an allele frequency (AF) < 0.05 in the 99Lives dataset, and also contains one non-coding variant that had the next highest CADD score after the *PEA15* variant.

| Gene | Transcript | HGVS DNA | HGVS Protein | Protein Change | 99 Lives AF | CADD | GERP | Cat Brain CPM |
|---|---|---|---|---|---|---|---|---|
| LY9 | XM_019822310.1 | c.654A>G | p.Pro218Pro | Synonymous | Absent | 0.0 | -5 | 3.3 |
| LY9 | XM_019822310.1 | c.478A>G | p.Met160Val | Missense | Absent | 0.0 | -7.9 | 3.3 |
| CD48 | XM_019822315.2 | c.695G>A | p.Arg232His | Missense | Absent | 6.6 | 2.1 | 3.2 |
| SLAMF1 | NM_001278826.1 | c.313C>T | p.Leu105Leu | Synonymous | 1.8% | NA* | 2.3 | 0.0 |
| PEA15 | XM_023247767.1 | c.176delA | p.Asn59fs | Frameshift | Absent | 29.7 | 5.5 | 1075.2 |
| ATP1A4 | XM_023247847.1 | c.1818C>T | p.Ala606Ala | Synonymous | Absent | No Liftover | | 1.0 |
| ATP1A2 | XM_019822324 | c.1044C>T | p.Arg348Arg | Synonymous | 4.4% | NA* | 2.8 | 1224.9 |
| ATP1A2-IGSF8 | NA (Intergenic) | FelCat8 n.65307418C>T | NA (Intergenic) | NA (Intergenic) | 0.1% | 21.1 | 2.4 | NA (Intergenic) |
| LOC101099681 | XM_003999605 | c.501G>A | p.Leu167Leu | Synonymous | 3.5% | No Liftover | | 0.0 |
| LOC101099681 | XM_003999605 | c.201G>A | p.Leu67Leu | Synonymous | 3.5% | 0.7 | -3.1 | 0.0 |

* The CADD scores for variants in SLAMF1 and ATP1A2 are 22.4 and 18.0, respectively. However, both variants predict non-synonymous changes in human (upon which the CADD scores are based), and synonymous changes in cat, and therefore likely overestimate potential deleteriousness.

In cats and other species, including humans, *PEA15* encodes a 15 kDa protein of 130 amino acids that is highly conserved across species (S4 Fig). The nucleotide deletion in *PEA15* lies towards the beginning of the second exon, and predicts a frameshift and truncated protein that lacks a critical ERK2 interaction domain (Fig 4A). Expression levels of *PEA15* as assessed

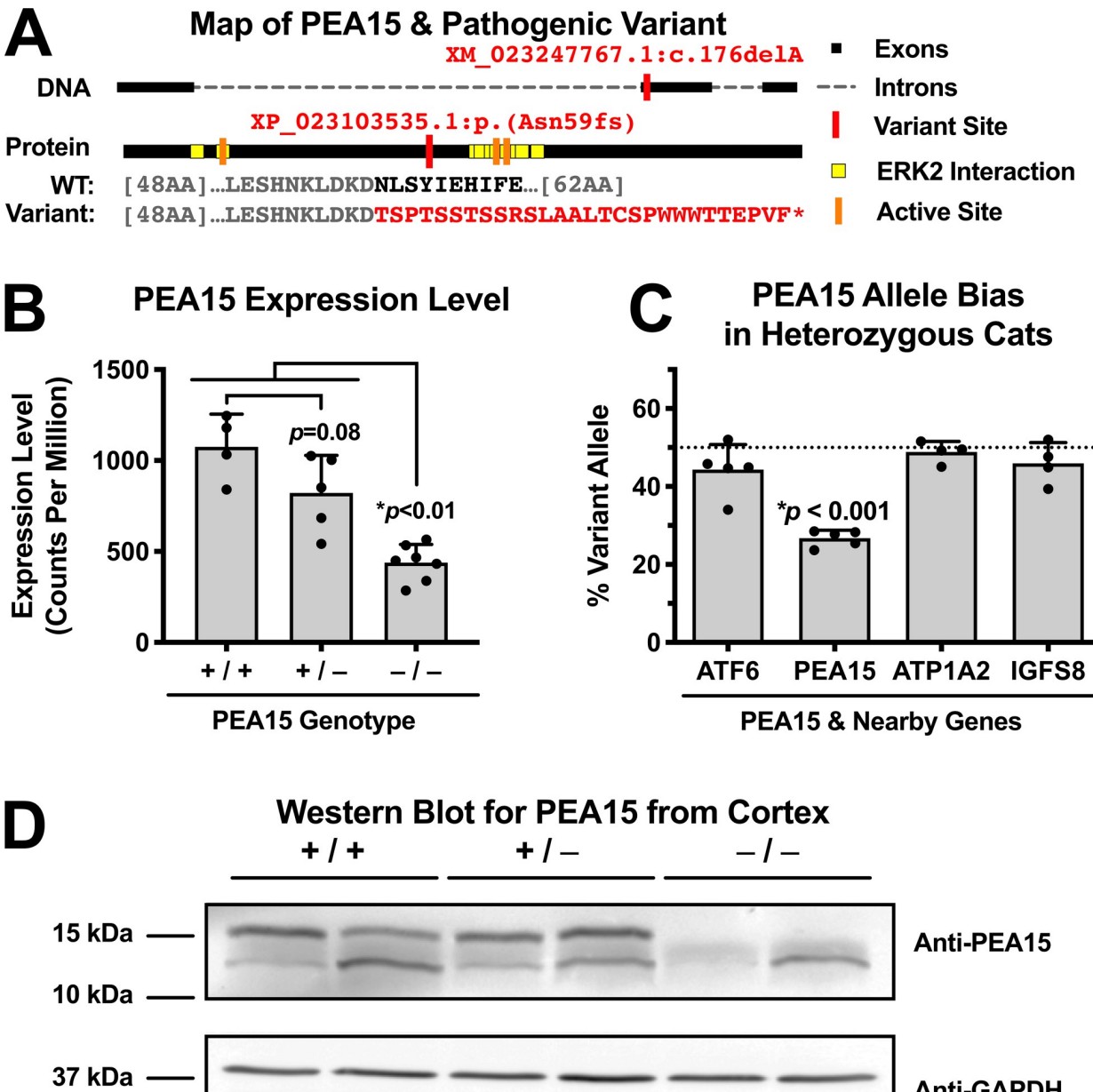

**Fig 4. The *PEA15* pathogenic variant introduces a premature termination codon, and PEA15 protein is absent in affected cats.** (A) Map of *PEA15* demonstrating the pathogenic variant location near the beginning of Exon 2. (B) Overall levels of *PEA15* transcripts measured by RNA-Seq are decreased in cats homozygous for the *PEA15* pathogenic variant (One-way ANOVA $^*p<0.0001$, $^*p<0.01$ by Tukey's *post hoc*). (C) Reads from the mutant *PEA15* allele in heterozygous cats are significantly reduced compared to non-mutant reads, while heterozygous variants in nearby genes do not exhibit allele bias (One-way ANOVA $^*p<0.0001$, $^*p<0.001$ by Tukey's *post hoc* vs. all of 3 nearby genes comparing the % variant reads per cat as the unit of comparison with 24 to >3,000 reads contributing to each % measurement for each gene). (D) PEA15 is absent from affected animals by western blot.

by RNA-seq (described below) from the cerebral cortex of adult cats reveal a reduction of 59% in homozygous affected animals, consistent with nonsense-mediated decay (NMD) (Fig 4B). In further support of NMD, in cats that are heterozygous for the 1.3 Mb critical region, we observed a significant allele bias in *PEA15* compared to other genes in the critical region (Fig 4C). We examined protein levels by Western blot with a polyclonal antibody against C-terminal amino acids 93–123 of Human PEA15; a 15 kDa band apparent in normal brain extracts was absent in brains from affected animals (Fig 4D).

## Neuropathology of cerebral dysgenesis

MAP2 immunohistochemically (IHC) stained sections (Fig 5A) were assessed along with Luxol fast blue (LFB) histochemical stained sections (Fig 5B) to evaluate changes in grey matter thickness and overall white matter area. Affected cats had variably decreased cortical grey matter thickness compared to age-matched controls (Fig 5A), and marked reduction of the white matter that comprises the corona radiata and internal capsule (Fig 5B). Consistent with MRI evaluation, there was an overall decrease in white matter which resulted in decreased area of LFB staining (Fig 5C).

In mammalian species, the cerebral cortex has a complex histological arrangement with six unique cortical layers that comprise different cell types: Layer I, known as the molecular, is closest to the cortical meningeal surface and contains predominately neuropil; Layers II and IV contain small stellate neurons; Layers III and V contain large pyramidal neurons and Layer VI contains fusiform neurons. These layers are most easily visualized in primates, and range from variably distinct in species such as the domestic cat to indistinct in rodents. Within the six layers, there are projection neurons, which mostly arise from Layers III and V and extend to form white matter (WM) tracts. Cortical layering in affected cats was more disorganized and variable in layer thickness compared to unaffected cats. Visually distinct cell populations like large pyramidal neurons were present (Fig 6A). The molecular layer was more prominent in LFB and GFAP stained sections owing to increased astrocyte density (Fig 6B). In unaffected cats, there were linear axonal projections that were oriented perpendicularly to the cortical meningeal surface. In cats with cerebral dysgenesis, perpendicular axonal orientation was reduced, with a denser and more haphazard orientation of neuronal processes as compared to unaffected cats (Fig 6C).

We assessed the density of astrocytes, cells of oligodendrocytic origin, and microglia by histomorphometry after immunostaining sections for GFAP, Olig-2, and IBA-1, respectively. The only significant change in affected cats was an increase in staining density for GFAP, due to increased numbers of grey matter astrocytes (Fig 7A–7E). There was no significant difference observed in Olig-2 staining (Fig 7F–7J), however it should be noted that Olig-2 detects both oligodendrocytic precursor cells, which can differentiate into neurons, astrocytes, and oligodendrocytes, as well as mature myelinating oligodendrocytes. No difference was detected in microglial density or morphology (Fig 7K–7O).

## RNA-seq analysis

To better understand the pathophysiologic mechanisms underlying cerebral dysgenesis, we analyzed RNA-seq data from cerebral cortex of adult homozygous mutant (n = 4), heterozygous (n = 3), and non-mutant (n = 3) cats. 61 genes exhibited significant (FDR <0.05) differential expression between homozygous mutant and non-mutant cats. Only 3 genes exhibited differential expression between heterozygous mutant and non-mutant cats. Given minimal differences by RNA-seq and that heterozygous cats are unaffected, we also performed a comparison of all 6 unaffected cats vs. homozygous mutant cats, revealing 25 differentially

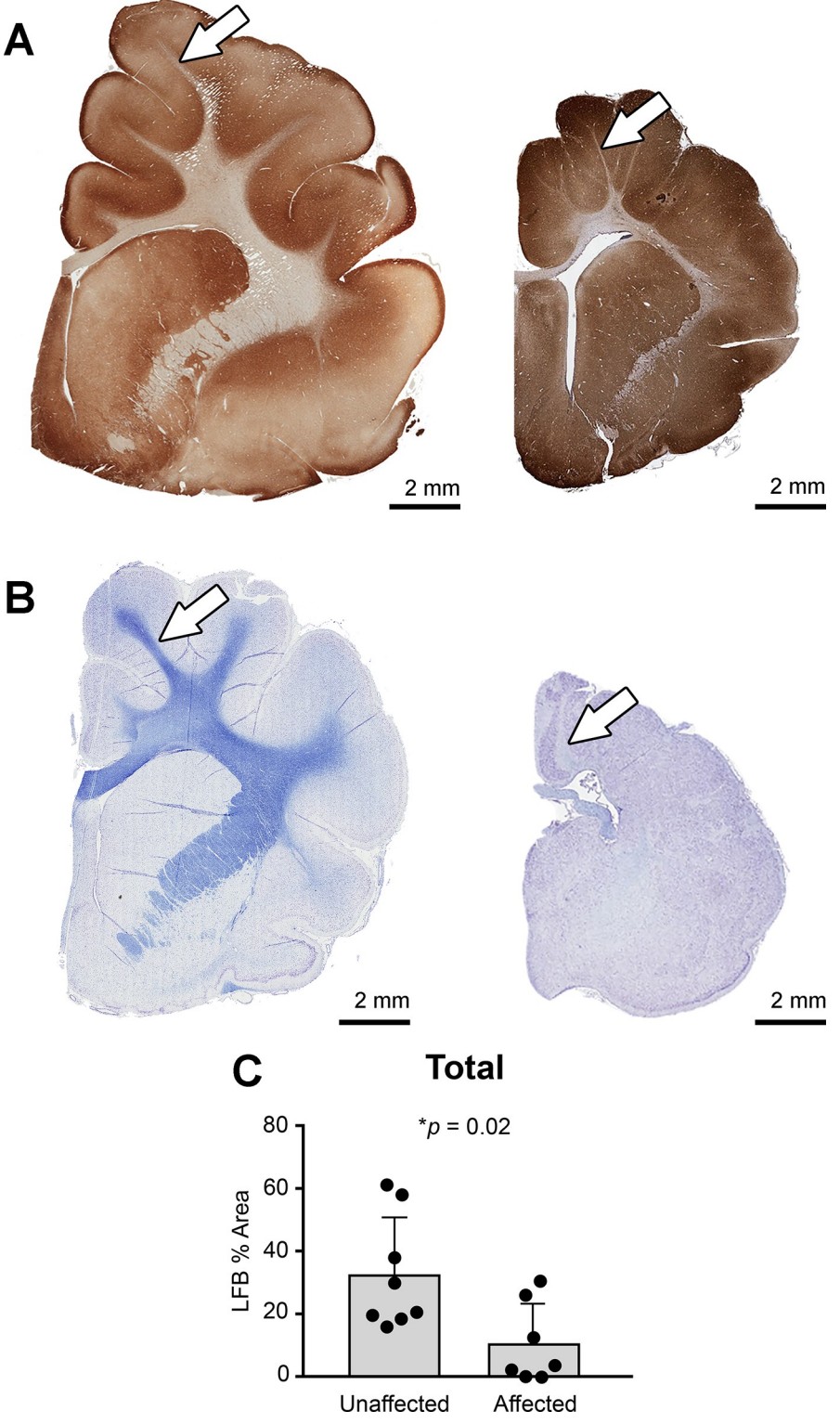

**Fig 5. Affected cats have a significant loss of white matter.** (**A**) Subgross sections of MAP2 stained neurons highlight the variable decrease in cortical thickness, and the reduced area of the corona radiata (arrows) and internal capsule. (**B**) Subgross sections of Luxol fast blue (LFB) stained for myelin indicates decreased white matter, (**C**) which is confirmed through quantification of LFB stained sections of the frontopareital region.

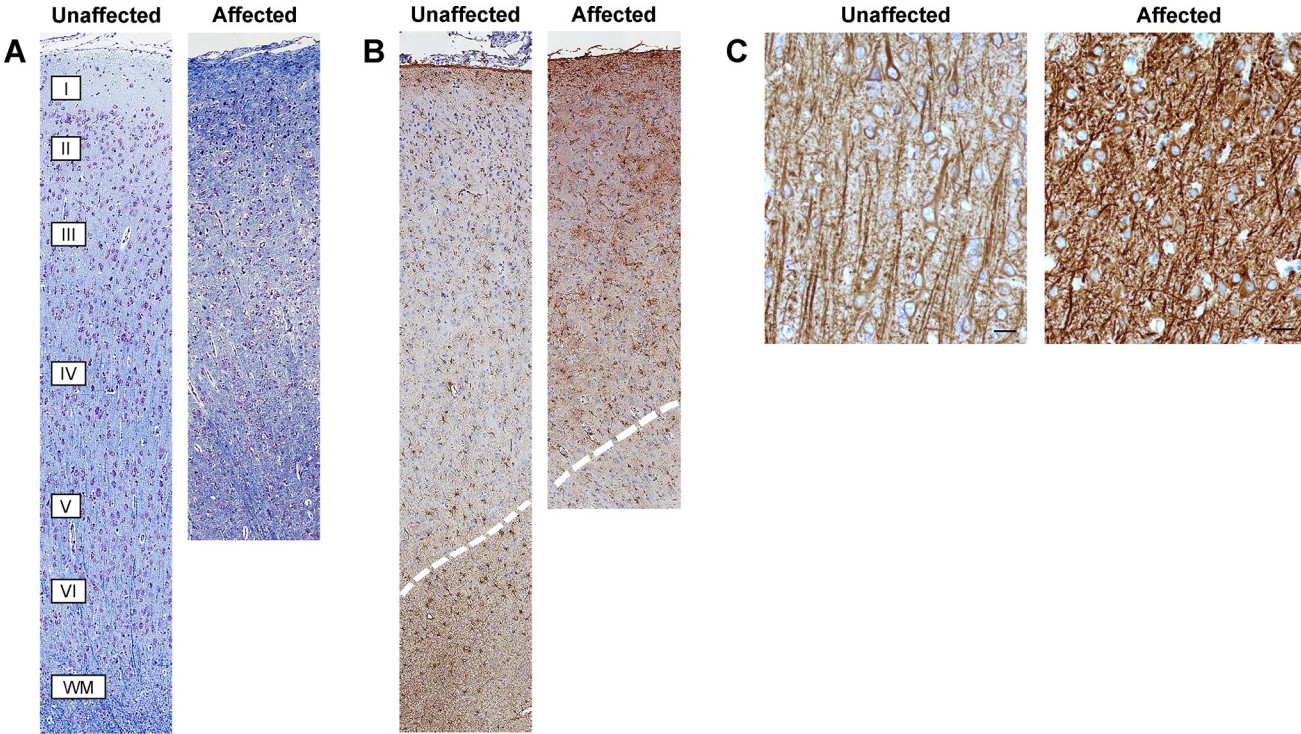

**Fig 6. Affected cats have loss of normal cerebral cortical layering, increased grey matter astrocytosis, and abnormal neuronal and axonal orientation.** (**A**) Photomicrographs of Luxol fast blue–Cresyl Echt Violet (LFB-CEV) stained sections depicting vertical columns in the parietal region of unaffected (left) and age-matched affected (right) cats. In unaffected cats, 6 cortical layers extend from below the meninges (beginning with layer I, molecular layer) to the white matter (WM). In affected cats, grey matter thickness and column morphology are altered with disorganized layering. (**B**) Photomicrographs of GFAP stained sections of vertical columns reveals that affected cats exhibit a relative astrocytosis. Dotted lines indicate separation of white matter and grey matter. (**C**) Photomicrographs from MAP2 stained sections taken at approximately layers IV and V. Unaffected cats have linear axonal projections oriented perpendicular to the cortical meningeal surface while affected cats lack axonal directionality (bar = 20uM). In all images, the meningeal edge is located at the top.

expressed (FDR <0.05) genes. The intersection of genes that were differentially expressed in both the all unaffected vs. homozygous mutant comparison and non-mutant vs. homozygous mutant comparison results in a list of 16 differentially expressed genes (Fig 8A; S4 Table), including 5 upregulated collagen genes (*COL6A5*, *COL4A5*, *COL3A1*, *COL1A1*, and *COL6A1*; Fig 8B).

These observations, together with the neuropathologic observations that suggested profound abnormalities in cerebral development and cellular organization, suggested that gene expression changes in cortex might be utilized to assess changes in cellular composition. To explore this further, we analyzed the RNA-seq data with cell type-specific deconvolution analysis [14], in which genes are organized and plotted according to their differential expression, the extent of enrichment in a specific cell type, and the nominal significance of that observation. As depicted in Fig 9, many transcripts enriched in oligodendrocyte precursor cells are overrepresented (Fig 9A), while many transcripts enriched in mature oligodendrocytes are underrepresented (Fig 9B). Transcripts for mature astrocytes (Fig 9C) and endothelial cells (Fig 9D) also exhibited a slight but significant enrichment, while no differences were observed for neurons or microglia (Fig 9E and 9F). Overall, these results are consistent with the histomorphometry data, and suggest a pathologic mechanism in which axonal disorganization, failure of gyrification, and microcephaly are secondary to expansion of astrocytes and reduction of myelin-associated cells.

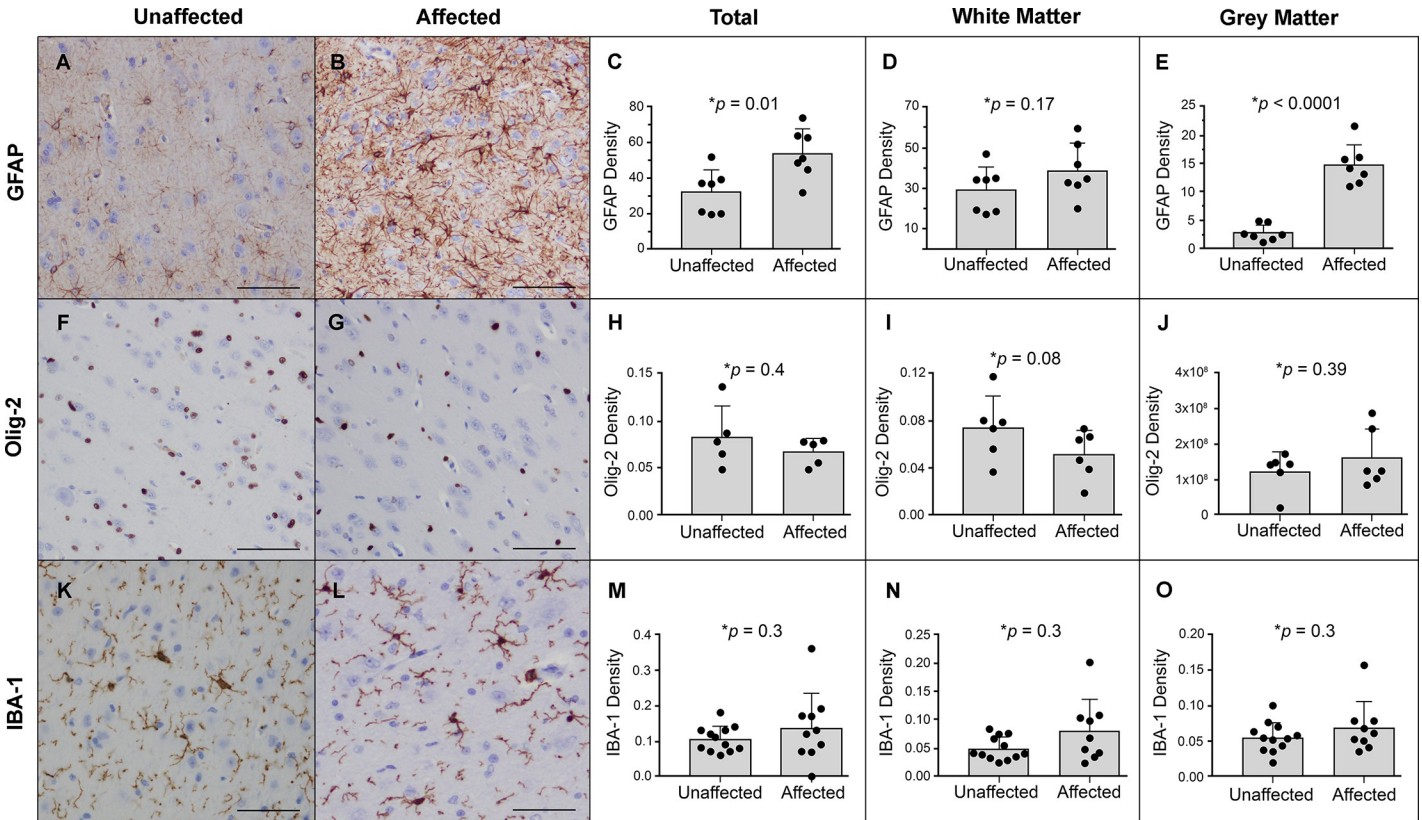

**Fig 7. Affected cats have significantly increased astrocyte density within the grey matter.** (**A-B**) GFAP immunohistochemistry (IHC) of grey matter indicates increased density of astrocytes. (**C-E**) Digital image-analysis algorithms measured a significant increase in GFAP stain density primarily in grey matter stain. (**F-J**) Olig-2 staining indicates no significant change in the density of oligodendrocytes in grey matter, though an insignificant decrease of ~30% was noted in white matter. (**K-L**) No significant change in the density or morphology of microglia was detected in affected cats. (**M-O**) Microglial density findings are confirmed on quantification of IBA-1 stain. (bar = 200um).

### Effects of PEA15 deficiency on apoptosis and cell proliferation in cats

Previous studies of *PEA15* in mice using a gene-targeted allele and a transgenic overexpression model demonstrated that it normally functions to suppress both TNFα-induced apoptosis [15] and cell proliferation during wound closure [16, 17]. We examined primary fibroblasts cultured from affected homozygous and non-mutant cats to investigate if those functions were conserved across species.

After treatment with TNFα, fibroblasts from affected animals exhibited reduced cell viability (Fig 9A) and increased caspase-8 activity (Fig 10B). Cell viability and caspase 8 activity did not change in the absence of TNFα treatment in fibroblasts from affected or unaffected animals. Exposure to FGFb for 72 hours yielded an ~ 2.5-fold increase in cell number in fibroblasts from unaffected animals, significantly elevated (p = 0.02) to ~3.5-fold in fibroblasts from affected animals (Fig 10C). Taken together, these results confirm that the effects of *PEA15* on apoptosis and cell proliferation are similar in cats and mice, and illustrate the functional impact of the *PEA15* pathogenic variant in cats with cerebral dysgenesis.

## Discussion

PEA15 was originally described more than 25 years ago as a substrate for protein kinase C that is associated with microtubules and highly enriched in astrocytes [18]. Subsequent studies

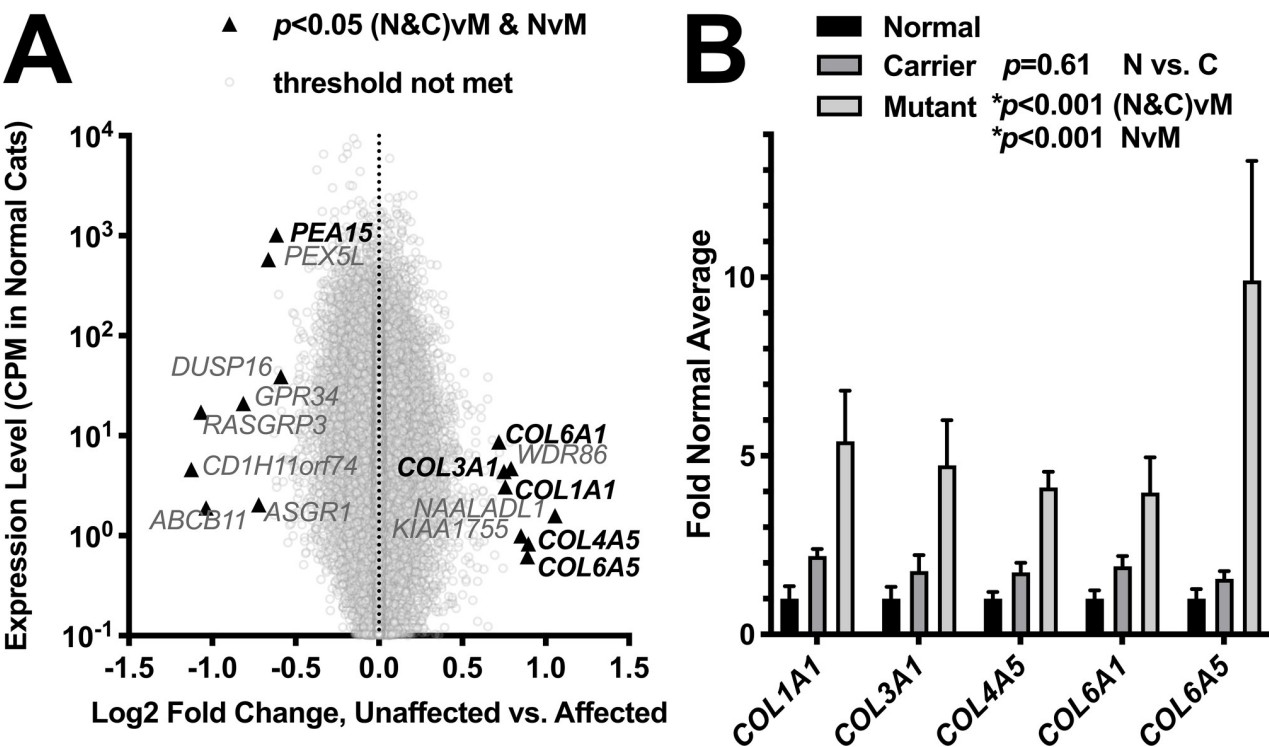

**Fig 8. Differential expression analysis.** (**A**) Log2-fold change vs. magnitude of gene expression for homozygous mutant (M–mutant) (n = 4) vs all unaffected (N&C–normal and carrier) (n = 6) (heterozygous mutant (C–carrier) (n = 3) and homozygous non-mutant (N–normal) (n = 3)). Genes with a significant difference for the strict criteria of significance in both homozygous mutant (n = 4) vs all unaffected (n = 6) and homozygous mutant (n = 4) vs homozygous non-mutant (n = 3) are labeled (triangles). Data was collected for an additional 6 animals, but excluded because of age, cause of death, or principal component analysis results (**Methods**; S5 Table; S5 Fig). (**B**) Quantitative changes in collagen gene expression for homozygous mutant, heterozygous, and homozygous non-mutant animals. No differences (p = 0.61, two-way ANOVA) were detected between normal and carrier cats.

indicated that *PEA15* is expressed at low levels in almost all tissues [19] but exhibits increased expression in the brain, particularly during late gestation and the early postnatal period [20]. In mouse fibroblasts, PEA15 was characterized as an adapter protein that regulates proliferation through cytoplasmic interaction with ERK1/2 [21–23], and receptor-mediated apoptosis through interaction with the Fas-associated death domain (FADD) [18]. Many studies on PEA15 have focused on a potential role in insulin resistance due to its increased expression in fibroblasts, skeletal muscle, and adipose tissue during states of insulin resistance (reviewed elsewhere [24]). *Pea15* knockout mice exhibited normal brain size and morphology [15], but displayed mild spatial and temporal learning deficits attributed to the potential role of PEA15 as a mediator of ERK-dependent spatial learning [25].

The *PEA15* frameshift pathogenic variant we identified in domestic cats is associated with a loss of steady state mRNA and immunodetectable protein, and fibroblasts homozygous for the pathogenic variant exhibit abnormalities in response to TNFα and FGFb that recapitulate what has been described previously in mice and mouse cells. In contrast to mutant mice, however, in which there are no obvious abnormalities in brain development, *PEA15* deficiency in cats is associated with extensive abnormalities in development and organization of the cerebral cortex that lead to a 45% decrease in overall brain weight, defective gyrification, expansion of astrocytes, and a loss of mature oligodendrocytes and white matter. Complete genomic sequence over the critical interval within which the cerebral dysgenesis pathogenic variant must lie identified no other plausible candidate variants except the *PEA15* frameshift

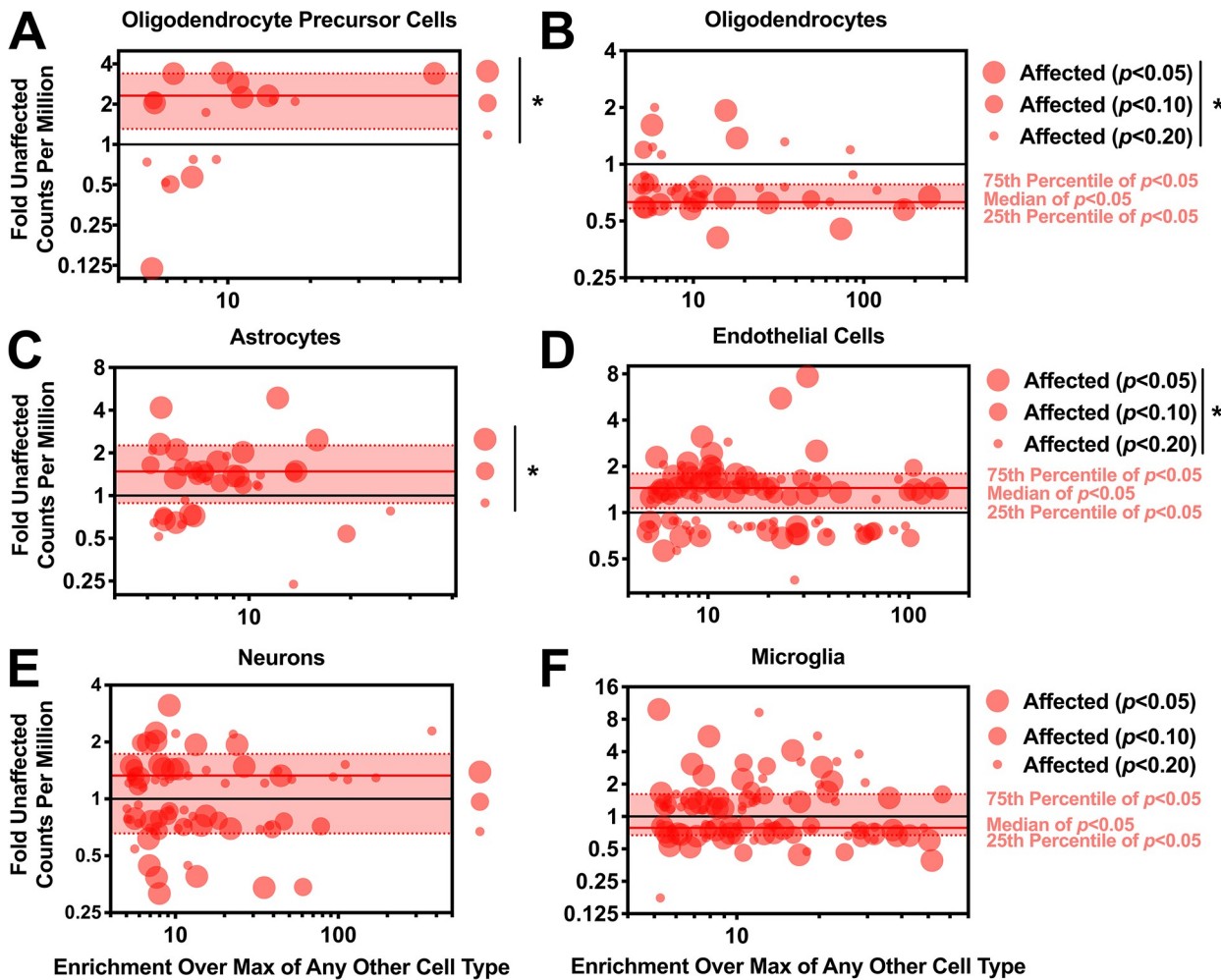

**Fig 9. Differences in levels of cell type–specific transcripts suggest altered cellular composition in the brains of affected cats.** (A) Oligodendrocyte precursor cell-specific transcripts are increased in affected cats (Two-way ANOVA *p<0.0001 genotype effect). (B) Oligodendrocyte-specific transcripts are decreased in affected cats (Two-way ANOVA *p<0.0001 genotype effect). (C) Astrocyte-specific transcripts are increased in affected cats (Two-way ANOVA *p<0.0001 genotype effect). (D) Endothelial cell-specific transcripts are increased in affected cats (Two-way ANOVA *p<0.0001 genotype effect). (E) Neuron–specific transcripts did not change significantly (Two-way ANOVA p = 0.60 genotype effect). (F) Microglia–specific transcripts did not change significantly (Two-way ANOVA p = 0.52 genotype effect).

pathogenic variant, and we conclude that *PEA15* deficiency is the likely cause of the neuropathologic abnormalities. Unlike many other domestic animals, application of gene editing technology to domestic cats faces a number of barriers and challenges, so it is not possible at present to further explore the pathogenicity of *PEA15* by generating new alleles, as might typically take place in laboratory mice. Nonetheless, there is a preponderance of evidence supporting a causal role for *PEA15* in cerebral dysgenesis, and we suggest that failure to observe a similar phenotype in mice has a simple explanation: the rodent brain is normally lissencephalic, and thus does not depend on extensive neuronal proliferation, expansion, and gyrification as it does in cats, dogs, and primates.

We propose a neuropathologic mechanism for the abnormalities described here whereby PEA15 normally serves to negatively regulate neuronal apoptosis and astrocyte proliferation (Fig 11, left), as has been demonstrated previously in mice. In the absence of PEA15, increased neuronal apoptosis and astrocyte proliferation leads to the production of excessive

**A**  **Cell Viability**

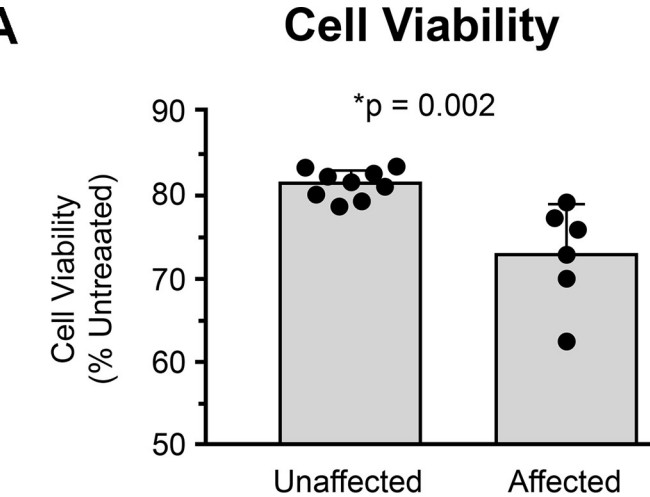

**B**  **Caspase-8 Activity**

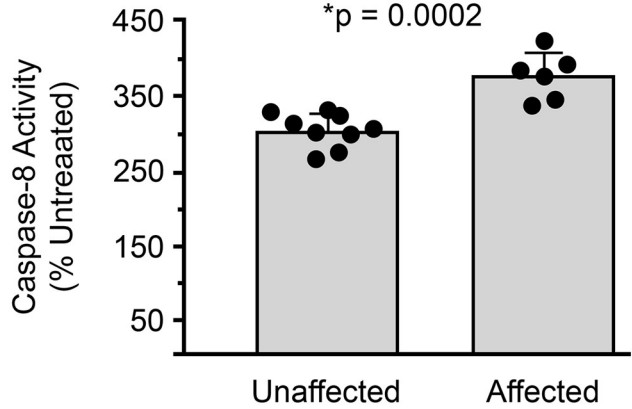

**C**  **FGFb Stimulated Proliferation**

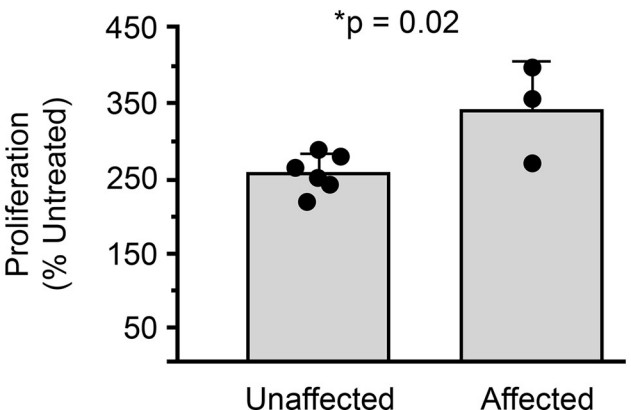

**Fig 10. Differences in cytokine-mediated apoptosis and proliferation in affected fibroblasts.** When treated with TNFα, there is (**A**) a significant decrease in cell viability and (**B**) a significant increase in caspase-8 activity of primary dermal fibroblasts from affected cats compared to unaffected cats. (**C**) When treated with 20ng/mL FGFb, proliferation is significantly increased in primary dermal fibroblasts from affected cats compared to unaffected cats. There was no significant difference of untreated cells for either genotype.

perineuronal nets, axonal disorganization and underdeveloped white matter tracts (Fig 11, right). Additional studies of *PEA15*-deficient cats should help determine at which phase of cortical development—neuroepithelial stem cell proliferation, neuronal stem cell migration, or synaptogenesis, apoptosis, and synaptic pruning—the postulated mechanisms are operative. We note, however, that the temporal pattern of *PEA15* expression, which peaks during late gestation and the early postnatal period, suggests a major role in the later stages of brain development, i.e. regulating neuronal apoptosis during synaptogenesis and synaptic pruning.

Although brain abnormalities in *PEA15*-deficient cats are striking, their gross appearance and behavior is not. Affected kittens were initially recognized due to a mild ataxia and were described by the husbandry staff as "shaky"; however, this gradually stabilized with age, and phenotype-based inference of disease status depends on an experienced clinician. Nonetheless, *PEA15* is under strong purifying selection—there is only a single amino acid substitution among mouse, cat, and human—and the gnomAD 2.1.1 [26] and TOPMed Bravo databases of human genome and exome data contains only 11 heterozygous loss-of-function variants (with none in the homozygous state) out of 198,527 non-overlapping individuals. Extrapolation suggests approximately five individuals on the planet with homozygous or compound heterozygous loss-of-function *PEA15* variants (assuming such a state would be consistent with life in humans), which may explain why it has not been previously recognized as a cause of human lissencephaly. In a recent summary of targeted sequencing studies for 17 genes in 811 patients with lissencephaly, a diagnostic rate of 81% was observed, in which $> 99\%$ of pathogenic variants were in *LIS1*, *DCX*, *TUBA1A*, or *DYNC1H1*, none of which are inherited in an autosomal recessive fashion.

In addition to identification of *PEA15* as a candidate gene for human lissencephaly, our work provides a new opportunity to investigate developmental mechanisms that underlie unique aspects of neurodevelopment in gyrencephalic species. Many of the processes disrupted by pathogenic variants in *LIS1*, *DCX*, *TUBA1A*, or *DYNC1H1* affect neuronal migration early in brain development, but, as noted above, the major role for *PEA15* in cortical development may occur later. As genomic tools for non-human animals continue to progress, careful clinical studies of companion and domestic animals are likely to lead to new insights into developmental and physiologic processes that are not present in conventional laboratory models.

## Materials and methods

### Study subjects and ethics statement

Animals included in this study are from the research colony at Auburn University College of Veterinary Medicine's Scott-Ritchey Research Center. Institutional Animal Care and Use Committee (IACUC) approval was obtained for all animal experiments. Animals evaluated in this study ranged in age from 1.2 months to 16 years and were evaluated in part based on videos that spanned a 20 year period. Based on breeding history, three adult cats within the colony were determined to be carriers. These cats along with three affected adults were also assessed by physical exam, including a complete neurological exam, serum biochemical analysis, complete blood count, MRI, and cerebrospinal fluid analysis. Cats were also evaluated for abnormalities associated with pathogenic variants in *GM2A* [11], and mucopolysaccharidosis VI

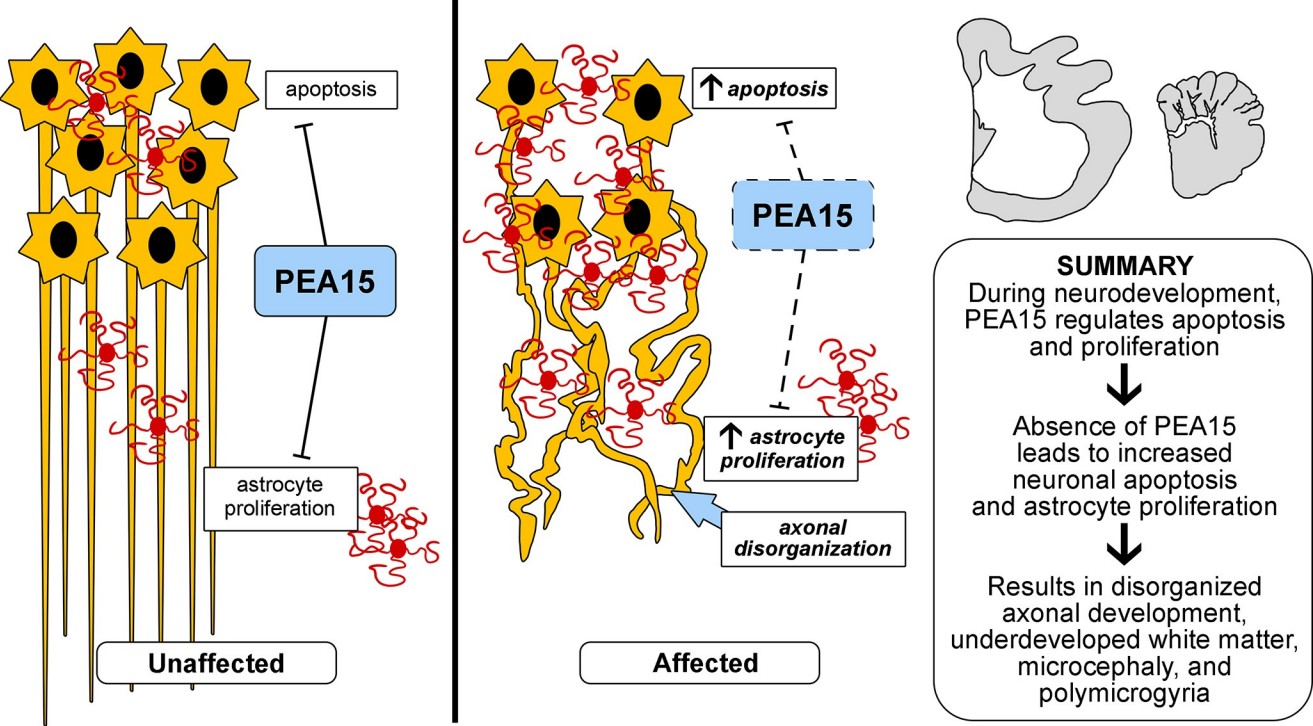

**Fig 11. Hypothesized mechanisms of PEA15 mediated cerebral dysgenesis in domestic cats.** Beginning in late gestation through the early post-natal period of normal animals, there is increased neuronal apoptosis during synaptic pruning. PEA15, which is normally expressed at this time in the brain, protects from excessive apoptosis of neurons and inhibits proliferation of stimulated astrocytes. Therefore, loss of PEA15 is expected to cause increased neuronal apoptosis and increased proliferation of astrocytes. Grey matter astrocytosis may be a direct response to the increased apoptosis or neurons (reactive astrocytosis), and/or and excessive proliferation due to loss of PEA15 function. Abundant astrocytes produce excessive extracellular matrix which can form perineuronal nets and cause a premature end of the critical period for synaptic plasticity. These changes in development result in disorganized axonal development and underdeveloped white matter tracts which manifest as cerebral dysgenesis.

(*MPSVI*) [12] which were independent of the phenotype observed here as described in the main text. Images of tissues depicted in Figs 1, 5, 6, and 7 are representative examples of a total of seven homozygous mutant and seven age-matched non-mutant animals that were examined by necropsy, and immunohistochemistry.

## Whole genome sequencing and RNA-seq–based genotyping

Whole genome sequencing was carried out on five animals from a single nuclear family for whom there was comprehensive information available regarding phenotype and breeding history: two affected individuals, their obligate carrier parents, and an obligate carrier sibling (based on affected offspring), as indicated in S1 Fig. For RNA-seq, we used tissue from two of the same animals used for WGS (a sibling pair—one affected, one obligate carrier), and 14 additional animals: six affected, four obligate carriers (based on pedigree information, S1 Fig.), and four unaffected and unrelated individuals with no known connection to the pedigree (S1 Table).

Genomic DNA was isolated from liver samples using a Qiagen DNeasy Blood & Tissue kit according to the manufacturer's instructions. DNA sequencing libraries were prepared for sequencing on the Illumina HiSeq X by the HudsonAlpha Genomic Services Lab by Covaris shearing, end repair, adapter ligation, and PCR using standard protocols. Library concentrations were normalized using KAPA qPCR prior to sequencing. After sequencing, adapters

were trimmed and FastQ files were quality checked using Trim Galore! 0.4.0 (a wrapper for cutadapt 1.8.1 [27] and FastQC 0.10.1). Initially, trimmed reads were aligned using bwa 0.7.12 [28] to the Felis Catus 8.0 build. When it became available, reads were also aligned to the Felis Catus 9.0 build. Most analyses were conducted with the Felis Catus 8.0 build because chain files for liftover from the 99 lives database were not available initially; however, analyses of candidate genes were carried out with the 9.0 build since this assembly has no gaps over the critical linkage region. Aligned reads were sorted and duplicates were marked with Picard tools 1.131. GATK 3.5.0 [29] was used for indel realignment, base quality recalibration, gVCF generation, and joint genotyping of genomic DNA. For RNA-seq data (described further below), variants were called using VarScan 2.3.6 [30]. For the zygosity analysis depicted in Fig 3A, we required genotyping calls from at least 11/14 animals, to allow for contribution from regions with less coverage in some cats. snpEff 4.1 [31] was used for annotation and filtering, and PROVEAN 1.1.5 [32] was used for missense damage prediction. For the *PEA15* frameshift, genotypes were confirmed by Sanger sequencing in 25 affected or obligate carrier animals.

## RNAseq and cell type deconvolution

Total RNA was extracted from the grey matter of the cerebral cortex of 16 cats (seven affected cats, five obligate carriers based on breeding records, S1 Fig, S1 Table, and four unaffected animals from a different pedigree that did not segregate the cerebral dysgenesis pathogenic variant) using the Qiagen RNeasy Lipid Tissue Mini Kit according to the manufacturer's instructions. Ages ranged from four months to six years for affected cats, 2–12 years for obligate carriers, and 2–5.5 years for unaffected cats. RNA was treated with TURBO DNase, and RIN scores were measured using a BioAnalyzer. Libraries were generated using polyA selection and an Illumina Nextera RNA-Seq protocol. Library concentrations were normalized using KAPA qPCR prior to sequencing. Libraries were sequenced with paired end 50 bp reads on an Illumina HiSeq 2500. Data from RNAseq was processed using aRNApipe [33], a python wrapper for several tools. Briefly, adapters were trimmed and FastQ files were quality checked using Trim Galore! 0.4.0 (a wrapper for cutadapt 1.8.1 [27] and FastQC 0.10.1). Reads were aligned with STAR 2.4.2a [34] to the Felis Catus 8.0 build and counted by gene with HTSeq 0.5.3 [35]. Count data were normalized and analyzed using DESeq2 [36] in R.

We collected RNA-seq data from 16 cats (7 homozygous mutant, 5 heterozygous, and 4 homozygous non-mutant). For the homozygous mutant vs. non-mutant comparison depicted in Fig 8A, 6 cats were excluded: 3 homozygous mutant kittens, 1 heterozygous mutant that died from a generalized seizure, and 1 heterozygous mutant and 1 homozygous non-mutant that were outliers on principal component analysis (S5 Fig). 10 cats (4 homozygous mutant, 3 heterozygous mutant, and 3 homozygous non-mutant) remained for analysis. Transcripts were considered for cell type deconvolution analysis if they were at least two fold enriched over any other cell type with a raw p value of less than 0.2 using relative abundance described previously [14].

## Amplicon sequencing

Amplicon sequencing was conducted for 96 cats, 91 with no previous genotype information along with one with genome sequencing and four with RNA-seq. Amplicons were chosen to span 26 variants spaced along the ~70 Mb interval that contained the candidate region on chromosome F1 as determined by zygosity analysis (Fig 3A), 4 additional variants space along the remainder of chromosome F1, and 20 variants from other chromosomes. Primers used are listed in S6 Table. After PCR with 1–2 primer pairs per reaction in 384-well plates, amplicons were pooled for each cat and barcoded for sequencing by ligating adapters to A-tails. Library

concentrations were normalized using KAPA qPCR prior to sequencing. Amplicons were sequenced using an Illumina MiSeq with 150 bp paired end reads. After sequencing, adapters were trimmed and FastQ files were quality checked using Trim Galore! 0.4.0 (a wrapper for cutadapt 1.8.1 [27] and FastQC 0.10.1). Trimmed reads were aligned using bwa 0.7.12 [28] to the Felis Catus 8.0 build. Picard tools 1.131 was used to sort, mark duplicates, and index. Platypus 0.8.1 [37] was used to call variants.

## Haplotyping, LOD score calculation, and other analysis & statistics

Merlin 1.1.2 [38] was used for haplotyping and LOD score calculation (S3 Table), according to a rare recessive model. Coverage was calculated using goleft indexcov 0.1.7 [39]. CNV analysis was conducted with Delly 0.7.8 [40]. Other packages used for standard processing of VCFs were vt [41], bcftools [42], and vcftools [43]. Liftover to hg38 for CADD [44] analysis was conducted with Crossmap 0.2.7 [45]. PEA15 conservation was assessed using a previously published sequence-to-structure-to-function workflow [46]. Other statistics were calculated in either R or Prism 7.

## Histological and immunohistochemical analysis

Histological and immunohistochemical (IHC) analyses were performed on cats that ranged in age from 1–8 months with age-matched controls. Luxol fast blue (LFB) staining was performed as previously described [47] both with and without Cresyl-Echt violet counterstain. Immuno-histochemical stains were performed using Dako automated immunostainer (Autostainer Link48, Dako-Agilent, Santa Clara, CA) using a low pH (6.1) antigen retrieval. IBA1 (Biocare Medical, CP290A; 1:100 dilution) and Olig2 (Abcam, EPR2678; 1:200 dilution) antibodies were incubated for 30 minutes. GFAP antibody (IR 52461–2 Dako-Agilent, Santa Clara, CA, no dilution) was incubated for 20 minutes. MAP2 antibody (Sigma-Aldrich, HM-2; 1:1000 dilution) was incubated for four hours. Detection was performed using the Dako EnVision HRP detection with DAB chromogen and hematoxylin counterstain. For quantification, slides were digitally scanned at 40x using an Aperio Scan Scope (Leica Biosystems, Buffalo Grove, IL, USA). Algorithms were written to quantify stained area of DAB for each individual IHC stain or LFB stain using Visiopharm quantitative digital histopathology software (Visiopharm, Hoersholm Denmark) and applied to all slides of an individual stain, except MAP2 as MAP2 had broad cytoplasmic staining that did not allow for individual cell density assessment.

## Magnetic resonance imaging

MRI data were acquired on a seven Tesla MAGNETOM scanner (Siemens Healthcare, Erlangen, Germany) from adult affected (n = 3), carrier (n = 2) and unaffected (n = 4) cats as previously described [48]. A 32-channel head coil (Nova Medical, Boston, Mass.) was used for all scans. Anatomical coronal images were acquired using 3D MPRAGE (Magnetization-PRepared RApid Gradient Echo) with 0.5mm isotropic resolution and TR/TE of 1910/2.5ms, followed by 2D axial T2 turbo spin echo (TSE) images with TR/TE of 5450/12ms and a resolution of (0.25x0.25x1) $mm^3$. MRI data were analyzed with EFilm 3.2 software (Merge Healthcare, Chicago).

## Immunoblot analysis

Samples of feline cerebral cortex (5–10 mg) were homogenized in 400 μl of RIPA buffer (PIERCE, ThermoFisher, Waltham, MA, USA) with a hand-held micro-pestle for 30 sec, followed by passage through a 25G needle and kept on ice for 20 min. After centrifugation at

16,200 ×$g$ for 30 min at 4˚C, the soluble fraction was transferred to a new tube and total protein concentration was determined by DC protein assay (Bio-Rad, Hercules, CA, USA). Following quantification, 33 μg of protein sample were mixed with 4× Laemmli sample buffer (Bio-Rad) containing 400mM Dithiothreitol. Protein fractions were separated with 18% sodium dodecyl sulfate polyacrylamide gel electrophoresis, and transferred to Odyssey® nitrocellulose membrane (Li-Cor, Lincoln, NE, USA), blocked in LI-COR Odyssey blocking buffer (Lincoln, NE) for 1 h, and incubated with rabbit polyclonal anti-PEA-15 (C-terminal amino acids 93–123 of Human PEA15) antibody ab135694 (Abcam, Cambridge, UK) at a concentration of 1:100 and anti-GAPDH antibody (MAB374, EMD Millipore, Burlington, MA, USA) at a concentration of 1:500. Secondary antibodies (1:15,000) were IRDye®680RD Goat anti-Rabbit IgG (H+L, Li-Cor) and IRDye®800CW Goat anti-Mouse IgG (H+L, Li-Cor), respectively. The fluorescent signal was detected using Odyssey® Infrared imaging system (Li-Cor).

## Primary skin fibroblast culture

Primary fibroblasts were established from feline skin samples. Dulbecco's modified eagle's medium (DMEM, Corning, NY) supplemented with 10% fetal bovine serum (FBS), penicillin (100 IU/ml), streptomycin (100μg/ml), and amphotericin B (0.25 μg/ml) was used as a standard growth media. Briefly, collected skin pieces were placed with the connective tissue in direct contact with the culture surface and cultured in growth media for 5 to 7 days until visible colonies formed. After removing skin pieces, cells were further cultured in growth media or stored in liquid nitrogen in freezing media containing 10% dimethyl sulfoxide.

## Cell viability, caspase-8 activity, and cell proliferation assay

Cell viability and caspase-8 activity were assessed in order to determine the susceptibility of primary fibroblasts to TNF-α induced apoptosis. Colorimetric Cell Viability Kit I (WST-8 reagent, PromoKine, Heidelberg, Germany) and Caspase-Glo® 8 Assay (Promega, Madison, WI) were used for cell viability assay and caspase-8 assay, respectively. In brief, cells were seeded in 96-well multiwell tissue culture plates at a density of 15,000 cells/90 uL/well. After 20 hours of incubation at 37˚C, cells were treated with 10 ng/ml purified recombinant human TNF-α (Peprotech, Rocky Hill, NJ) prepared in growth media containing 10 μg/ml of Actinomycin D (MP Biomedicals, Solon, OH). Two identical plates were prepared for each experiment. Following 3 hours of TNF-α treatment, one plate was equilibrated to room temperature for 15 min, caspase-8 assay substrate was added and the luminescent signal was evaluated by Infinite M200 microplate reader (Tecan, Mannedorf, Switzerland) after 30 minutes of room temperature incubation. The other plate was cultured for 6 hours post TNF-α treatment. WST-8 reagent was added to each well and the plate was incubated at 37˚C for 1.5 hours. Then, the absorbance at 450 nm was measured by Infinite M200 microplate reader. Cell proliferation was assessed in primary fibroblasts as percent response to fibroblast growth factor-b (FGFb) relative to untreated cells from each individual. Cells were seeded in 6-well multiwell tissue culture plates at a density of 90,000 cells/2 mL/well. After 20 hours of incubation at 37˚C, cells were treated with 20 ng/ml purified FGFb (Peprotech, Rocky Hill, NJ) prepared in growth media. Following 72 hours of FGFb treatment, cell were washed once with PBS(-) and dethatched with 0.25% trypsin-EDTA (Corning). The total cell number in each well was determined using trypan blue dye exclusion on a hemocytometer.

## Supporting information

**S1 Table. Table of cats from the GM2A and MPSVI breeding colonies where the cerebral dysgenesis pathogenic variant was identified.** Affected status is denoted by color with

unaffected in white, obligate carriers based on breeding in grey, and affected cats in black.
(PDF)

**S2 Table. Cerebrospinal Fluid Analysis.** Cerebrospinal fluid protein concentration and cell counts in affected and carrier cats with a reference interval based on normal cats that was established by the Auburn University small animal teaching hospital.
(PDF)

**S3 Table. Haplotyping and LOD Score Calculation.** Merlin 1.1.2 was used for haplotyping and LOD score calculation. LOD score was calculated using parametric linkage analysis assuming a rare recessive model. Each marker is considered independently, equivalent to a theta value of zero. Note that, while we focused genotyping on chromosome F1 based on homozygosity mapping, we selectively genotyped a handful of markers on other chromosomes.
(PDF)

**S4 Table. Variants from Fig 3A.** These 10 variants met initial zygosity filtering criteria from WGS and RNA-seq data.
(PDF)

**S5 Table. RNA-seq counts and differentially expressed genes.** See excel file. Summary statistics are listed for each gene, and counts per million are listed for each cat, with exclusion criteria for cats not included in summary statistics noted.
(XLSX)

**S6 Table. Primer pairs for amplicon sequencing.** Note that multiple primer pairs were included for the top nominated region to maximize genotyping success and increase coverage. The *PEA15* frameshift site is bolded.
(PDF)

**S1 Fig. Pedigree.** Phenotype is denoted by color and indicated on the pedigree with unaffected in white, obligate carriers with a central dot, and homozygous mutant cats in black.
(PDF)

**S2 Fig. Cerebrospinal fluid enzyme activity.** (**A**) Changes in aspartate amino transferase and (**B**) lactate dehydrogenase enzyme activity in adult cats (n = 3) from the cerebral dysgenesis cohort.
(PDF)

**S3 Fig. FelCat9 coverage.** (**A**) When averaged over 10,000 base pair windows, coverage stays above 30x for all cats across the window linked to the phenotype. (**B**) When averaged over 1,000 base pair windows, only a few regions dip below 20x coverage. (**C**) Regions with less than 20x coverage in all 5 cats. Note that regions are either in repetitive intronic or intergenic regions.
(PDF)

**S4 Fig. PEA15 Conservation.** (**A**) Conservation of PEA15 gene sequence was performed using the open reading frames from 150 species. Scores at each codon were assessed, where 100% conservation corresponds to a score of 1, and this score also receives the addition of 0 if dN-dS of the site is below the mean, addition of 0.25 for sites with values above the mean to 1 standard deviation above the mean, addition of 0.5 for sites greater than 1 standard deviation but below 2 standard deviations, and addition of 1 for sites greater than 2 standard deviations. Therefore, a score of 2 is maximal suggesting an amino acid that is 100% conserved with

codon wobble indicative of a high selection rate at the position. (**B**) Conservation values were placed on a 21-codon sliding window (combining values for 10 codons before and after each position) to identify conserved motifs within the gene. (**C**) Model of PEA15 protein with a structural z-score of 0.12 (assessed with YASARA2 knowledge-based force field) suggesting accurate predictions of fold space. Colors are based on 150 species alignments fed into Con-Surf.
(PDF)

**S5 Fig. Principal component analysis (PCA) of RNA-seq data from cat cortex.** Out of 16 original cats, 1 cat was excluded from further analysis because of death by grand mal seizure, evident by PCA. 3 cats were excluded from further analysis because they were kittens to avoid detection of developmental false positive signals in differential expression analysis (1 kitten was also an outlier by PCA). 2 cats were excluded on the basis of being clear outliers by PCA. 10 cats (4 homozygous mutant, 3 heterozygous mutant, and 3 homozygous non-mutant) remained for analysis.
(PDF)

# Acknowledgments

We thank Joseph Benito for technical assistance in generating RNA-seq libraries, Corneliu Henegar for assistance with computational scripts, Bandon Brunson for initial review of the histology, Edward Morrison for his review of the histopathology and guidance, Nancy Merner for her help with the pedigree, Matt Miller for reading and revising the manuscript, and Nancy Morrison for technical support in generating primary feline fibroblasts. 99 Lives Consortium members that contributed to the 99 Lives analysis used in this manuscript include: Leslie A. Lyons[1], Danielle Aberdein[2], Paulo C. Alves[3,4], Holly C. Beale[5], Adam R. Boyko[6], Jeffrey A. Brockman[7], Marta G. Castelhano[8], Patricia P. Chan[5], N. Matthew Ellinwood[9], Jonathan E. Fogle[10], Dorian J. Garrick[2,9], Christopher R. Helps[11], Marjo K. Hytönen[12], Maria Kaukonen[12], Emilie Leclerc[13], Tosso Leeb[14], Hannes Lohi[12], Maria Longeri[15], Richard Malik[16], Michael J. Montague[17], John S. Munday[2], William J. Murphy[18], Niels C. Pedersen[19], Max F. Rothschild[9], Joshua A. Stern[19], William F. Swanson[20], Karen A. Terio[21], Rory J. Todhunter[8], Yu Ueda[19], Wesley C. Warren[17], Elizabeth A. Wilcox[8], Julia H. Wildschutte[22], Barbara Gandolfi[1]

[1]Department of Veterinary Medicine and Surgery, College of Veterinary Medicine, University of Missouri, Columbia, Missouri, 65211, United States of America

[2]School of Veterinary Science, Massey University, Palmerston North 4442, New Zealand

[3]CIBIO/InBIO, Centro de Investigação em Biodiversidade e Recursos Genéticos/InBIO Associate Lab & Faculdade de Ciências, Universidade do Porto, Campus e Vairão, 4485–661 Vila do Conde, Portugal

[4]Wildlife Biology Program, University of Montana, Missoula, Montana, 59812, United States of America

[5]Maverix Biomics, Inc., San Mateo, California, 94402, United States of America

[6]Department of Biomedical Sciences, College of Veterinary Medicine, Cornell University, Ithaca, New York, 14853, United States of America

[7]Hill's Pet Nutrition Inc., Topeka, KS 66601, United States of America

[8]Department of Clinical Sciences, College of Veterinary Medicine, Cornell University, Ithaca, New York, 14853, United States of America

[9]Department of Animal Science, College of Agriculture and Life Sciences, Iowa State University, Ames, Iowa, 50011, United States of America

[10]College of Veterinary Medicine, North Carolina State University, Raleigh, NC 27607, United States of America

[11]Langford Veterinary Services, University of Bristol, Langford, Bristol, BS40 5DU, United Kingdom

[12]Department of Veterinary Biosciences and Research Programs Unit, Molecular Neurology, University of Helsinki and Folkhälsan Research Center, Helsinki 00014, Finland

[13]Diana Pet food, Inc. SPF–ZA du Gohelis, 56250 Elven, France

[14]Vetsuisse Faculty, Institute of Genetics, University of Bern, 3001 Bern, Switzerland

[15]Dipartimento di Medicina Veterinaria, University of Milan, 20122 Milan, Italy

[16]Centre for Veterinary Education, University of Sydney, Sydney, NSW, 2006, Australia

[17]The McDonnell Genome Institute, Washington University School of Medicine, St. Louis, Missouri, 63108, United States of America

[18]Department of Veterinary Integrative Biosciences, College of Veterinary Medicine, Texas A&M University, College Station, Texas, 77845, United States of America

[19]Department of Medicine and Epidemiology, School of Veterinary Medicine, University of California at Davis, Davis, California, 95616, United States of America

[20]Center for Conservation and Research of Endangered Wildlife (CREW), Cincinnati Zoo & Botanical Garden, Cincinnati, Ohio, 45220, United States of America

[21]Zoological Pathology Program, University of Illinois, Brookfield, IL 60513, United States of America

[22]Bowling Green State University, Department of Biological Sciences, Bowling Green, OH 43403, United States of America

## Author Contributions

**Conceptualization:** Emily C. Graff, J. Nicholas Cochran, Christopher B. Kaelin, Kenneth Day, Nancy R. Cox, Gregory S. Barsh, Douglas R. Martin.

**Data curation:** Emily C. Graff, J. Nicholas Cochran, Nancy R. Cox.

**Formal analysis:** Emily C. Graff, J. Nicholas Cochran.

**Funding acquisition:** Gregory S. Barsh.

**Investigation:** Emily C. Graff, J. Nicholas Cochran, Christopher B. Kaelin, Kenneth Day, Heather L. Gray-Edwards, Rie Watanabe, Jey W. Koehler, Rebecca A. Falgoust, Jeremy W. Prokop.

**Methodology:** Emily C. Graff, J. Nicholas Cochran, Christopher B. Kaelin, Gregory S. Barsh.

**Project administration:** Richard M. Myers, Gregory S. Barsh, Douglas R. Martin.

**Resources:** Gregory S. Barsh, Douglas R. Martin.

**Software:** J. Nicholas Cochran.

**Supervision:** Christopher B. Kaelin, Gregory S. Barsh.

**Validation:** Emily C. Graff, J. Nicholas Cochran.

**Visualization:** Emily C. Graff, J. Nicholas Cochran.

**Writing – original draft:** Emily C. Graff, J. Nicholas Cochran, Gregory S. Barsh.

**Writing – review & editing:** Emily C. Graff, J. Nicholas Cochran, Gregory S. Barsh.

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
