## [Decision Letter · Decision Letter 0]

30 Mar 2020

Dear Dr Barsh,

Thank you very much for submitting your Research Article entitled 'PEA15 loss of function and defective cerebral development in the domestic cat' to PLOS Genetics. Your manuscript was fully evaluated at the editorial level and by independent peer reviewers. The reviewers appreciated the attention to an important topic but identified some aspects of the manuscript that should be improved.

We therefore ask you to modify the manuscript according to the review recommendations before we can consider your manuscript for acceptance. Your revisions should address the specific points made by each reviewer.

[LINK]

Yours sincerely,

Hannes Lohi, PhD

Guest Editor

PLOS Genetics

Gregory P. Copenhaver

Editor-in-Chief

PLOS Genetics

Reviewer's Responses to Questions

**Comments to the Authors:**

Reviewer #1: The authors report the discovery of a recessive mutation in cats that leads to severe alterations in brain structure and function that produces a variety of associated traits for which abundant phenotypic data is provided at behavioral, anatomical and microanatomic levels. Interestingly, the affected feline gene is PEA15 in which knockout mice models show no structural alterations. In vitro studies of fibroblasts support increased apoptosis and proliferation of other cell types and implicate PEA15 function in these pathways.

The identification of the likely causal variant from WGS and RNAseq as presented within Fig 3 and the text is complete and well done, as is the sequence, expression and protein quantitation alteration presented in Fig 4.

The abundance of supporting data for the phenotypic effects is far beyond many/most domestic animal model genetic investigations. I am not an expert in any aspect of the neuroanatomy or function, but am very much inclined to trust the author’s conclusions. Further the in vitro data indicating PEA15 is involved in intracellular signaling pathways is quite sufficient as well. The discussion is also well-presented and succinct.

My only comment is in regards to the 25 cats from the colony had the cerebral dysgenesis phenotype in which genotyping for 2 lysosomal storage disease mutations showed that the CB locus is not co-segregating. In and around line 102 please clearly state how many of the 25 CD affected cats were homozygous for the GM2A and the MPSVI wild type alleles. And how many were homozygous for both wild type alleles.

Reviewer #2: This submission impressively illustrates the current possibilities of WGS-based precision medicine in cats. The success rate of WGS-based identification of rare disease-causing genes is difficult to determine because about two-thirds of studied human diseases discover too many credible candidate variants and in the remaining one-third there are none. It is obviously of advantage to study rare conditions in purebred companion animals like cat. This in combination with the aspect of potential model usage for human research instead of the standard rodent model highlights the value of this work. The observed brain anomaly is specific to mammals with gyrifcation that is not present in mice or rats.

As a non-expert in both feline medicine and neuroanatomy I would like to restrict my comments on the positional cloning which is somehow representing the core of this submission. The genome-wide mapping is difficult to read, and although the fine-mapping on F1 (Figure 3B) looks very convincing it could be presented in the supplement. Also the data illustrating the results of the additionally performed linkage analysis could be moved to the supplement (Figure 3C+D) as it looks somehow redundant. The confirmation of the two mapping approaches are written in the text, I guess that is sufficient in the light of the entire length of the manuscript with many nice figures.

WGS data filtering was based on the assumption of autosomal recessive inheritance but focussed only on protein-changing variants. Please specify “all variants” mentioned in line 151 with detailed numbers, e.g. in supplementary table. Furthermore the chosen mapping approach based on homozygosity is not clearly presented as the resolution of Figure 3A is poor. A table reporting all shared ROHs across the entire genome would be more informative. Furthermore the possibility of biallelic pathogenic variants was not considered (compound heterozygosity). During the fine-mapping based on amplicon-resequencing it looks like that only protein-changing variants were considered (lines 155-158 and 164-169), some more details would be nice to have. Why all non-coding variants were ruled out at this stage? In addition, it is not quite clear why only the 1.3Mb region on F1 was further studied. Why it was allowed to assume that the disease-causing variant is fully absent from the 99 Lives variant catalogue?

Minor comments:

Please strictly adhere to the published guidelines of sequence variant nomenclature (http://varnomen.hgvs.org/) e.g. lines 85-86 of line 165. Avoid talking about mutations, maybe variant would be more appropriated. Finally, please also adhere to the guidelines for the interpretation of sequence variants (https://www.nature.com/articles/gim201530). Accordingly the LoF variant could be classified as “pathogenic”. The variant details should also be presented in Figure 4A. Within the first section of the results it would good to refer to the OMIA database (https://omia.org/) to refer to the tested variants in GM2A and MPSVI.

Reviewer #3: PEA15 LOSS OF FUNCTION AND DEFECTIVE CEREBRAL DEVELOPMENT IN THE DOMESTIC CAT

Graff and coworkers have identified a loss-of function mutation in the feline PEA15 gene that appears to cause defective cerebral development in the domestic cat. The affected cats exhibited impaired cerebral cortical expansion and folding as well as microcephaly. In particular, the cerebral cortex showed abnormalities. To investigate to genetic causes underlying the cerebral dysgenesis, the authors performed whole genome sequencing of eight affected cats and six obligate carriers and used conditional filtering for identified genetic variants under the assumption of an autosomal recessive mode of inheritance. RNA-seq was also done from cerebral cortex from four affected cats, three obligate carriers and three “non-mutant” cats.

GENERAL COMMENTS

This is an impressive study with a thorough analysis of a large amount of data and the resulsts will significantly contribute to the biological understanding of the observed abnormalities in the development of the cerebral cortex. A significant portion of the manuscript is focused on the pathology of the condition. This is very important when defining a condition. However, for most readers of PLOS Genetics, the pathological and histological results may be difficult to follow. A more explanatory description would help the reader with the interpretation of some of the histology. For example, Figure 6 includes six layers plus the white matter (WM). For an untrained eye and for non-experts in neuro-pathology, it will be difficult to observe the six layers and to know what constitutes the layers. I would therefore urge the authors to consider being more descriptive and explain the histology and the pathology for non-pathologist.

The genetic analyses including whole-genome sequencing, the RNA-seq as well as the bioinformatics is very well described and the conclusions are well supported. It is particularly fortunate that the authors have access to a sufficient number of individuals and tissues to perform a robust experimental set up and analyses.

In short, the conclusions of this study, including that the identified deletion in PEA15 is likely responsible for cerebral dysgenesis in the investigated domestic cats, is well supported by the Graff and co-workers.

MINOR REVISIONS AND COMMENTS

Please check the text under Table 1. “Of the two variants that change coding sequence, the missense variant in LY9 is not predicted to be damaging, and LY9 is not expressed in brain.”

In the results section under “RNA-seq analysis” the sentence below with “unremarkable phenotypically” sounds somewhat strange. It is clear what you mean but please check the language. “Given minimal differences by RNA-seq and that heterozygous mutant cats are unremarkable phenotypically, we also performed a comparison of all 6 unaffected cats vs. homozygous mutant cats, revealing 25 differentially expressed (FDR <0.05) genes.”

Table S3. Why is it question marks after the heading “On Figure?” and “PEA15 Variant?”? Please clarify.

**Have all data underlying the figures and results presented in the manuscript been provided?**

Reviewer #1: Yes

Reviewer #2: Yes

Reviewer #3: Yes

PLOS authors have the option to publish the peer review history of their article (what does this mean?). If published, this will include your full peer review and any attached files.

Reviewer #1: No

Reviewer #2: No

Reviewer #3: No

---

## [Editor Report · Decision Letter 1]

10 Jun 2020

Dear Dr Barsh,

We are pleased to inform you that your manuscript entitled "PEA15 loss of function and defective cerebral development in the domestic cat" has been editorially accepted for publication in PLOS Genetics. Congratulations!

Yours sincerely,

Hannes Lohi, PhD

Guest Editor

PLOS Genetics

Gregory P. Copenhaver

Editor-in-Chief

PLOS Genetics

**Data Deposition**

http://datadryad.org/submit?journalID=pgenetics&manu=PGENETICS-D-20-00206R1

**Press Queries**

---

## [Editor Report · Acceptance letter]

21 Sep 2020

PGENETICS-D-20-00206R1 

PEA15 loss of function and defective cerebral development in the domestic cat 

Dear Dr Barsh, 

We are pleased to inform you that your manuscript entitled "PEA15 loss of function and defective cerebral development in the domestic cat" has been formally accepted for publication in PLOS Genetics! Your manuscript is now with our production department and you will be notified of the publication date in due course.

With kind regards,

Kaitlin Butler

PLOS Genetics

On behalf of:
